# AHF: An Automatic and Universal Image Preprocessing Algorithm for Circular-Coded Targets Identification in Close-Range Photogrammetry under Complex Illumination Conditions

Hang Shang and Changying Liu *

College of Instrumentation and Electrical Engineering, Jilin University, Changchun 130061, China; shangxing20@mails.jlu.edu.cn
* Correspondence: liuchangy@jlu.edu.cn

**Abstract:** In close-range photogrammetry, circular-coded targets (CCTs) are a reliable method to solve the issue of image correspondence. Currently, the identification methods for CCTs are very mature, but complex illumination conditions are still a key factor restricting identification. This article proposes an adaptive homomorphic filtering (AHF) algorithm to solve this issue, utilizing homomorphic filtering (HF) to eliminate the influence of uneven illumination. However, HF parameters vary with different lighting types. We use a genetic algorithm (GA) to carry out global optimization and take the identification result as the objective function to realize automatic parameter adjustment. This is different from the optimization strategy of traditional adaptive image enhancement methods, so the most significant advantage of the proposed algorithm lies in its automation and universality, i.e., users only need to input photos without considering the type of lighting conditions. As a preprocessing algorithm, we conducted experiments combining advanced commercial photogrammetric software and traditional identification methods, respectively. We cast stripe- and lattice-structured light to create complex lighting conditions, including uneven lighting, dense shadow areas, and elliptical light spots. Experiments showed that our algorithm significantly improves the robustness and accuracy of CCT identification methods under complex lighting conditions. Given the perfect performance under stripe-structured light, this algorithm can provide a new idea for the fusion of close-range photogrammetry and structured light. This algorithm helps to improve the quality and accuracy of photogrammetry and even helps to improve the decision making and planning process of photogrammetry.

**Keywords:** circular-coded target; close-range photogrammetry; tie points extraction; homomorphic filtering; complex illumination; genetic algorithm; structured light

## 1. Introduction

Close-range photogrammetry has many flourishing and diverse applications in many fields, such as archaeology [1], physiology [2], architecture [3], agriculture [4], and industry [5]. One of the main tasks of close-range photogrammetry is identifying homologous points in multiple images, commonly called the correspondence issue [6–10]. Due to the continuous development of various photogrammetric software, image correspondence issues can be automatically resolved—the software searches for features uniquely identified in multiple images. Most software uses general structure from motion (SFM) technology [4], primarily finding dense textures on objects, such as text, wood grain, facial features, and other patterns. There is also the use of shape from shading (SFS) technology [11], which further enriches data through lighting and shading cues. However, artificial coded targets are more prevalent in high-precision close-range photogrammetry, which is a very accurate method. Due to unique identity information, coded targets are distinguishable,

easy to identify, and quickly match automatically, and always serve as a tie point between images with overlapping regions. They are more suitable for weak texture scenes and even for reflective, transparent, and other featureless surfaces [12], e.g., in the industrial field, geometric feature detection and large-scale industrial surfaces lack significant texture information. Using self-adhesive paper to paste coded targets onto the surface of an object is a commonly used low-cost measurement method [13].

Many scholars have researched the design style of coded targets, which can be roughly divided into three categories [14]: point-distributed [15–18], circular [19–23], and color-coded targets [24,25]. Although color can be used to improve the reliability of identification, color-coded targets cannot be effective in all situations. Shorits et al. [26] elaborated on the limitations of color-coded targets. Figure 1 shows two representative coded targets in close-range photogrammetry. Figure 1a shows the coded targets of GSI's V-STARS system (here-after called GCT) [15]. Point-distributed coded targets have the advantages of large coding capacity and high identification accuracy. However, as reported by Tushev et al. [17,27], GSI did not disclose the identification method of GCT due to trade secrets. Recently, Wang et al. [28] attempted to develop an identification method for GCTs, but the results were inferior to the identification effect of V-STARS. Under the illumination of a point light source, elliptical light spots will be formed on the image, affecting the identification of GCTs. Therefore, in identifying coded targets under the illumination of a point light source, CCTs have more advantages in identification than GCTs.

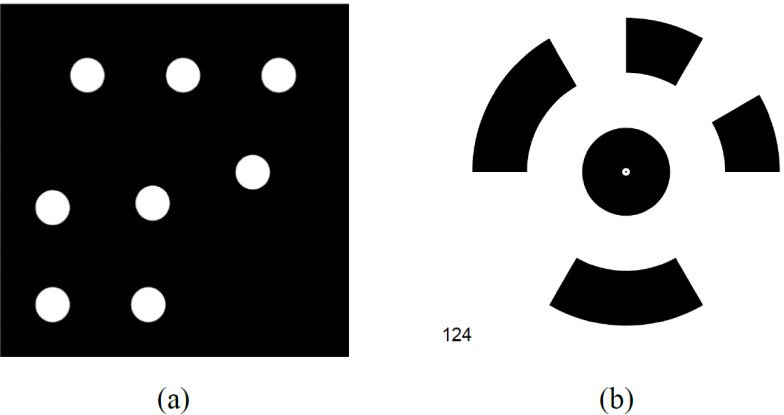

(a)  (b)

**Figure 1.** Two representative coded targets: (**a**) GCT. (**b**) Schneider CCT.

Schneider CCTs [19] (see Figure 1b) are typical examples of CCTs. The CCT has a simple and compact structure with the advantage of affine invariance. They are applied to commercial close-range photogrammetric software Agisoft Metashape [29]. The Schneider CCT is public [28]. Therefore, we can continue to study the latest identification methods based on previous studies, which will inevitably lead to increasingly robust identification.

Many methods have been proposed for CCT identification, and the existing techniques almost share a common strategy. Generally, there are two gradual steps: central positioning and decoding. Firstly, the CCT is filtered from the image background. Traditional methods rely on multiple judgment criteria [30–34], such as ellipse circumference, ellipse roundness, gray-level criteria, location criteria, etc. An object detection network has been used based on deep learning for filtering [23,35]. Then, the central positioning method includes ellipse fitting based on image edges [36,37], the Hough transforms approach [38], and the gray-weighted centroid method based on gray image intensity [39]. Finally, decoding is performed by searching for a coded band through the center point for scanning and decoding. Generally, the coded band is directly located based on the geometric structure of the CCT, while Liu et al. [14] and Kong et al. [40] searched for the coded band through the grayscale averaging method, which is helpful for identification under flat viewing angles. Decoding operations are based on grayscale gradient methods [41–46].

The prerequisite for successful identification methods is a high-quality image. In identifying CCTs, the most basic Otsu threshold segmentation method [47] and Canny edge detection method [48] are affected by illumination conditions. On this basis, the grayscale centroid method for central positioning, the grayscale averaging method for searching coded bands, and the grayscale gradient method for decoding are all very sensitive to the internal grayscale distribution, and the grayscale intensity distribution is very quickly affected by illumination conditions. This results in inaccurate positioning, ineffective decoding, or undetectable CCTs. However, using metric [49] or high-resolution consumer-grade digital cameras [50] in close-range photogrammetry tasks cannot effectively address the impact of complex illumination on image quality.

Up to now, only a few scholars have focused on this challenging issue. Tushev et al. [17,27] proposed a method for identifying coded targets under adverse lighting conditions. Still, their approach is to change the material of the coded target to a reflective material and add a camera with a flash to take photos. We do not want to increase the cost of materials, but we seek algorithmic improvements. Liu et al. [14] adopted an adaptive brightness adjustment method to solve the identification of camera overexposure and underexposure situations. However, these issues are relatively easy to solve. Our previous work tried to combine close-range photogrammetry and structured light technology for three-dimensional (3D) surface topography measurements of large objects [5]. When projecting stripe- or lattice-structured light on the surface of the measured object, such illumination conditions are very complex, covering almost all complex illumination that can be encountered, such as point lights, multiple lights, uneven illumination, and shadow areas. Even if the advanced commercial photogrammetric software Agisoft Metashape is used to identify CCTs, the results are still not ideal.

Dey [51] reviewed image processing algorithms for handling uneven illumination, such as retinex, quotient image, adaptive histogram equalization, illumination normalization, logarithmic transforms, homomorphic filtering (HF), etc. In these image processing algorithms, HF can optimize the impact of lighting on images and stretch contrast rather than enhance features. Dong et al. [52] combined HF with the gray-weighted centroid method to improve the accuracy of circular marker positioning under uneven illumination conditions. It has been verified that HF can be used for high-precision measurement. It is also shown that HF can improve the image quality of uneven illumination without affecting measurement accuracy. However, the effect of image enhancement depends on the parameters of HF, and the parameters required for various types of illumination conditions are also different. Dong et al. [52] determined various parameters of HF through simulation experiments. Currently, the adaptive HF algorithms proposed are all based on the characteristics of the image itself. For example, Venkatappareddy et al. [53] determined adaptive parameters by calculating the pixel average and variance of the input image; Fan et al. [54] determined through experiments that one of the HF parameters has a significant impact on laser images and derived an empirical formula for this parameter based on the average gray level of the image; Gamini et al. [55] used a genetic algorithm to optimize the parameters of HF and represented the fitness function by combining the sum of entropy, edge intensity, and the number of edge pixels of the image.

For the identification of CCTs, the actual situation is more complex. Locating accurate and relatively complete coded bands is necessary to ensure correct decoding. When the image's contrast is too high, or the edge details of the image are enhanced too much, the incomplete coded band will affect identification. Therefore, only relying on the characteristics of the image itself to study adaptive algorithms cannot adapt to various illumination conditions. For the identification of CCTs, the parameter values of HF are also irregular. Up to now, there has yet to be a unified method for identifying CCTs. Although the design strategies of the identification methods are generally similar, there are still differences in the design concepts of many scholars. Therefore, adaptive algorithms based on the characteristics of the image itself cannot be used in conjunction with any CCT identification

method, which is not universal. There are apparent limitations if we use it in combination with a specific identification method.

This paper proposes an AHF as a preprocessing technique for CCT identification. It can solve the impact of complex illumination and combine it with any CCT identification method. It does not need to consider the characteristics of the input photo. It uses a GA to optimize HF parameters using the CCT's identification results as the objective function. Through continuous iteration, it seeks optimal identification results. Therefore, the algorithm proposed in this article can be universal and fully automated. So far, this article is the first to offer a general algorithm for CCT identification under complex illumination conditions.

## 2. Proposed Algorithm

In this section, we introduce our proposed algorithm. The entire algorithm flow is shown in Figure 2, divided into three steps. The first step is to determine whether the procedure needs to be executed. The second step is HF, which is used to solve the impact of complex illumination. The third step is to optimize the process to achieve universal and automated functions.

To save computational resources, we determine whether it is necessary to execute this program. First, the photo is input into the CCT identification method. Then, whether the identified number of CCTs $n$ equals the total number of targets $N$ in the picture is determined. If similar, the identification result is directly output. Otherwise, executing this procedure means that the image needs to be enhanced.

Here are some suggestions for setting the total target number $N$:

1. We can enter the total number of pasted CCTs in advance with prior knowledge;
2. If unknown, a general object detection network [56] can identify CCTs. The recognition result is treated as ground truth, i.e., the number of targets $N$.

### 2.1. Homomorphic Filtering (HF)

Fundamentally, no matter how complex illumination is, it can be considered as the uneven illumination of the local description of the image. The details of the parts of the image that correspond to low brightness are difficult to distinguish. The primary purpose of enhancement methods based on HF is to eliminate the impact of uneven illumination. Its basic principle is to compress the illumination component in the original image, expand the reflection component, and enhance the image's details based on the image's illuminance–reflection model [52]. The specific algorithm is as follows, in five steps.

Step 1: Take the logarithm of the image function.

$$z(x,y) = \ln\left[f(x,y)\right] = \ln\left[i(x,y)\right] + \ln\left[r(x,y)\right] \tag{1}$$

where $f(x,y)$ represents the grayscale value of the image at the location $(x,y)$. $i(x,y)$ and $r(x,y)$ are illumination and reflection functions, respectively.

Step 2: Perform Fourier transform.

$$Z(u,v) = \mathcal{F}[z(x,y)] = \mathcal{F}[\ln[i(x,y)]] + \mathcal{F}[\ln[r(x,y)]] = I(u,v) + R(u,v) \tag{2}$$

Step 3: Perform high-pass filtering processing.

$$S(u,v) = H(u,v)Z(u,v) = H(u,v)I(u,v) + H(u,v)R(u,v) \tag{3}$$

The low-frequency component primarily corresponds to the illumination component, and the high-frequency component mainly corresponds to the reflection component. Therefore, it is only necessary to design a suitable high-pass filter transfer function $H(u,v)$ to have different effects on low-frequency and high-frequency components to reduce the illumination component and expand the reflection component. As in Equation (4), a modified Gaussian high-pass filter transfer function is used.

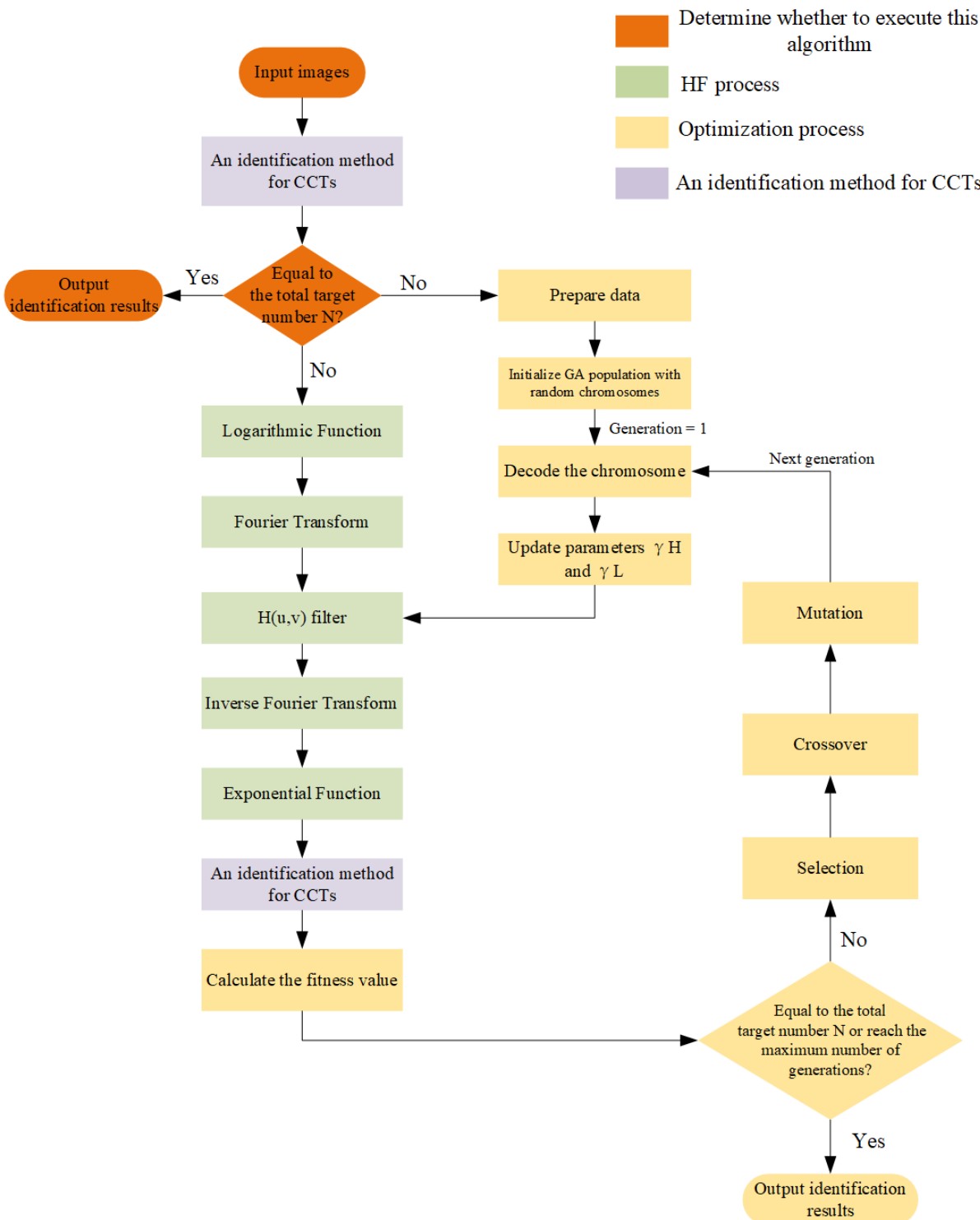

**Figure 2.** Proposed algorithm flow framework.

$$H(u,v) = (\gamma H - \gamma L)[1 - e^{-c(\frac{D(u,v)}{D_0})^2}] + \gamma L \tag{4}$$

where $\gamma H$ and $\gamma L$ represent the filtering characteristics of high-frequency and low-frequency components, respectively; when $\gamma H > 1$, $0 < \gamma L < 1$, indicating that the illumination component is suppressed and the reflection component is enhanced. $D(u,v)$ is the distance

from $(u, v)$ the origin of the centered Fourier transform, as shown in Equation (5). $D_0$ is the cutoff frequency. $c$ is the attenuation factor.

$$D(u, v) = \sqrt{(u - u_0)^2 - (v - v_0)^2} \tag{5}$$

Step 4: Perform inverse Fourier transform.

$$s(u, v) = \mathcal{F}^{-1}[S(u, v)] = \mathcal{F}^{-1}[H(u, v)I(u, v)] + \mathcal{F}^{-1}[H(u, v)R(u, v)] = i'(x, y) + r'(x, y) \tag{6}$$

Step 5: Perform exponential transformation to obtain the HF image.

$$g(x, y) = e^{s(u,v)} = e^{i'}(x, y)e^{r'}(x, y) = i_h(x, y)r_h(x, y) \tag{7}$$

where $i_h(x, y)$ and $r_h(x, y)$ are the illumination and reflection components after HF.

*2.2. Optimization Process*

2.2.1. Optimization Problem

According to the introduction in Section 2.1, the high-pass filter transfer function $H(u, v)$ is the main factor affecting the HF effect, and its parameters include the following:

- $\gamma H$: Used to adjust the high-frequency components in the filtered image, and increasing $\gamma H$ results in an increase in the high-frequency components in the picture;
- $\gamma L$: Used to adjust the low-frequency components in the filtered image, and increasing $\gamma L$ causes the low-frequency components in the image to expand;
- Cutoff frequency $D_0$: The frequency at which the filter stops attenuating the input signal. In HF, the cutoff frequency determines the degree of detail retention in the image. A lower cutoff frequency preserves more low-frequency components, resulting in a smoother picture; Higher cutoff frequencies keep more high-frequency components, resulting in more explicit photos;
- Attenuation factor $c$: determines the rate at which the filter attenuation frequency exceeds the cutoff frequency. Higher attenuation factors can lead to a steeper transfer function slope, resulting in a clearer image, but may introduce more noise. A lower attenuation factor produces a smoother appearance but may lose some details, usually between $\gamma H$ and $\gamma L$.

Choosing the optimal HF parameters is a trade-off between filtering performance and computational complexity. $\gamma H$ and $\gamma L$ play a significant role, and to reduce the number of parameter combinations arranged, the cutoff frequency $D_0$ can be selected by calculating the median of $D(u, v)$ [48,50]. The attenuation factor $c$ can be set to

$$c = \gamma H - \gamma L \tag{8}$$

Therefore, the optimization problem of CCT identification under complex illumination conditions can be transformed into the essential parameters $\gamma H$ and $\gamma L$ for optimizing HF. We address the optimization process through the following steps.

Step 1: We input the parameters into HF's high-pass filter transfer function.

Step 2: The image processed through HF is sent to the CCT identification method to identify the number of targets $n$.

Step 3: We judge whether the identified number of targets $n$ equals the total number $N$. Therefore, we summarize this complex optimization problem into a straightforward objective function, such as Equation (9), which uses the identification results of CCTs as the objective function, cleverly avoiding the issue that optimization algorithms based on the characteristics of the image itself do not have universality.

$$minimize|N - n| \tag{9}$$

where $|\cdot|$ represents taking an absolute value, because in the identification method of CCTs, there may be false positives, which can lead to $n > N$.

Step 4: We obtain the target number $n$ of the minimum quantity difference and corresponding HF parameters through optimization.

Step 5: We output the optimal CCT identification result.

### 2.2.2. GA Modeling

Due to the existence of trade secrets, various commercial photogrammetric software may not be open-source methods for identifying CCTs. We also do not need to know how they are identified, which is the advantage of our preprocessing technology's versatility. Therefore, the internal model relationship of the optimization process has yet to be discovered. It requires finding an optimal solution within a reasonable computational time or a solution close to the answer (i.e., the total target number). One of the most effective artificial intelligence methods is a GA miming natural evolutionary processes [57]. A GA has all the characteristics of intelligent search [58] and a global random search algorithm [59]. Therefore, we use a GA to optimize HF parameters for CCT identification. In this section, we describe the deployment optimization model in detail.

1. Fitness Function
   The only requirement for using a GA is that the model must be able to determine the value of the objective function, and the objective function used to select offspring should be a non-negative growth function of individual quality [60]. In our optimization problem, we use Equation (9) as the objective function, and our goal is to minimize Equation (9). To adapt to a GA, we use its reciprocal as the fitness function. The fitness value is shown in Equation (10).

$$Fitness = \frac{1}{|N - n| + \delta} \tag{10}$$

   where $\delta$ is an infinitesimal value, preventing an error when the denominator is 0.

2. Decision Variables
   The decision variables are the parameters $\gamma H$ and $\gamma L$ of HF. The value range of $\gamma H$ is [1,4], and the quantization during optimization is 0.1. The value range of $\gamma L$ is [0, 1], and the quantization during optimization is 0.01. Due to the different value ranges and coding accuracy of the two decision variables, the multiparameter cascade coding method is adopted. Each parameter is first binary-coded to obtain a substring; then, these substrings are connected in a specific order to form a chromosome (individual). Each chromosome corresponds to an optimized set of variable values.

3. GA Structure
   As shown in Figure 2, our optimization process includes various modules of the GA. In the following, the functions and parameter configurations of each module are described. Table 1 lists the specifications of the GA used in this algorithm.

   - Preparing data: In the GA, several parameters must be determined to converge and save computational resources quickly. We first use the four parameters of HF, i.e., $\gamma H$, $\gamma L$, $C$, and $D_0$. At the same time, the population size, maximum generation, crossover probability, and mutation probability are fixed.

   - Initialization: Initialize the GA population with random chromosomes. Different identification methods, illumination conditions, and HF parameters are others.The calculation cost of combined permutations of HF parameters is tremendous. For example, the value range of $\gamma H$ is [1, 4], and the quantization during optimization is 0.1. The value range of $\gamma L$ is [0, 1], and the quantization during optimization is 0.01. There are 3000 combined permutations in total. Due to the Fourier transform required after each combination permutation, choosing a simple brute-force optimization algorithm would waste many computational resources. We randomly select $\gamma H$ and $\gamma L$ and use Equation (7) to determine the parameter $c$, while the median value of the calculated $D(u, v)$ determines $D_0$.

   - Fitness function calculation: We update the parameters $\gamma H$ and $\gamma L$ by decoding the chromosomes for individuals. The image processed by HF is fed into a

CCT identification method, and the fitness function value is calculated using the number of targets $n$ identified by Equation (10).

- Optimization stop criteria: "Is it equal to the total target number $N$ or reaches the maximum generation?" is used as the judgment module. If it is equal to the total target number $N$, the identification result is directly output; if the maximum generation is reached, the ultimate fitness member of the final GA is selected, and the identification result is output. If none is achieved, proceed to the next step, such as "Selection", "Crossover", and "Mutation".

**Table 1.** The specifications of the applied genetic algorithm.

| Parameter | Value or Method |
|---|---|
| Population size | 50 (chromosome) |
| Selection | Tournament |
| Crossover | Single-point |
| The probability of crossover | 0.8 |
| The probability of mutation | 0.1 |
| Stopping criterion | "Lack of progress over 50 successive generations" OR "The total target number N was reached" |

## 3. Experiments and Analysis

### 3.1. Experimental Setup

As shown in Figure 3, 54 standard specifications of CCTs were pasted on the wall. A Canon EOS 850D SLR camera was used for image acquisition and was equipped with an EF-S 18–55 mm f/4–5.6 lens. The Optoma EH415e projector was used to create complex lighting conditions. The specifications of the camera and projector are shown in Table 2.

**Table 2.** The specifications of the camera and projector.

| Device | Parameter | Value |
|---|---|---|
| Canon EOS 850D SLR Camera | Resolution | 3984 × 2656 pixels |
| | Focal length | 55 mm |
| Optoma EH415e Projector | Resolution | 1920 × 1080 pixels |
| | Horizontal scanning frequency | 15.375–91.146 KHz |
| | Vertical scanning frequency | 24–85 Hz |
| | Brightness | 4200 lux |

We experimented with advanced photogrammetric software Agisoft Metashape (AM) and the latest publicly published traditional CCT identification method (TI). Due to this study's lack of relevant literature, we cannot compare it with other algorithms, so we choose AM and TI as the control groups. Our adaptive homomorphic filtering algorithm (AHF) is used for image preprocessing. Comparative experiments combine AM (referred to as AHF-AM) and TI (referred to as AHF-TI) to verify the effectiveness and accuracy of the proposed algorithm.

AM is a tool for three-dimensional reconstruction, visualization, measurement, and mapping tasks and is currently a highly competitive photogrammetric software on the market [29,40]. The workflow of TI [33] is first to filter out CCTs based on criteria such as ellipse circumference and roundness and then use ellipse fitting to locate the center. Then, we calculate the gray gradient of the coded band to obtain the center angle of each coded band segment for decoding. It is worth noting that the purpose of the experimental comparison is to verify whether the proposed algorithm effectively improves the performance of the CCT identification method under complex illumination conditions. At the same time, it is also shown that this algorithm can be used as a preprocessing technique in combination with any CCT identification method.

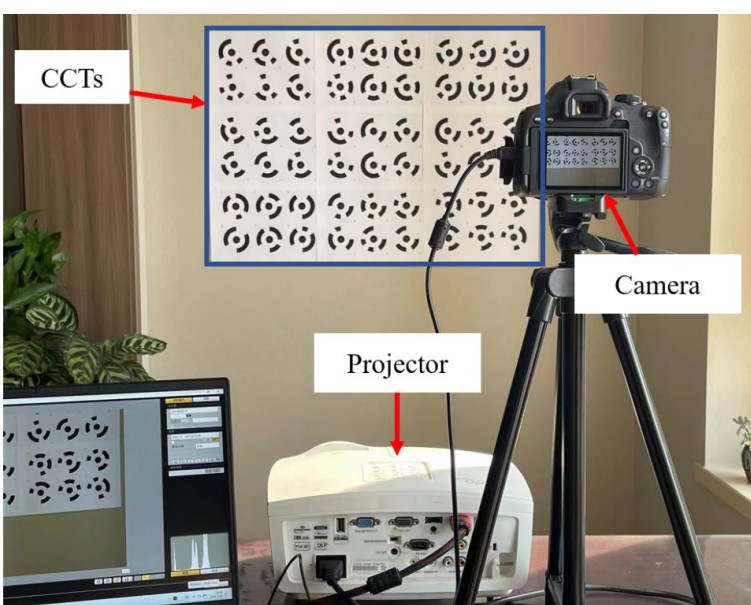

**Figure 3.** Experimental setup.

We chose a projector to project stripe- and lattice-structured light on the image to create complex lighting conditions. The following complex types of lighting conditions may be involved:

- Shadows: Casting structured light may produce shadows on the CCT surface, making some areas of low light intensity or completely dim. The presence of shadows can cause CCTs to be invisible and unrecognizable;
- Uneven illumination: This uneven lighting condition may cause changes in the brightness of the CCT surface, affecting the extraction and decoding of CCTs;
- Light spot: Densely projectedlattice-structured light, which will form an oval spot when imaging and may damage the imaging structure of the CCT.

Such lighting conditions are challenging, including the poor lighting environment encountered in close-range photogrammetry applications. Therefore, to thoroughly verify this algorithm's performance, three groups of experiments of lighting conditions are set, as shown in Figure 4: normal lighting, striped structured light, and lattice-structured light. AM, TI, and AHF-AM, AHF-TI all use the same image, and each group obtains 50 photos for comprehensive evaluation.

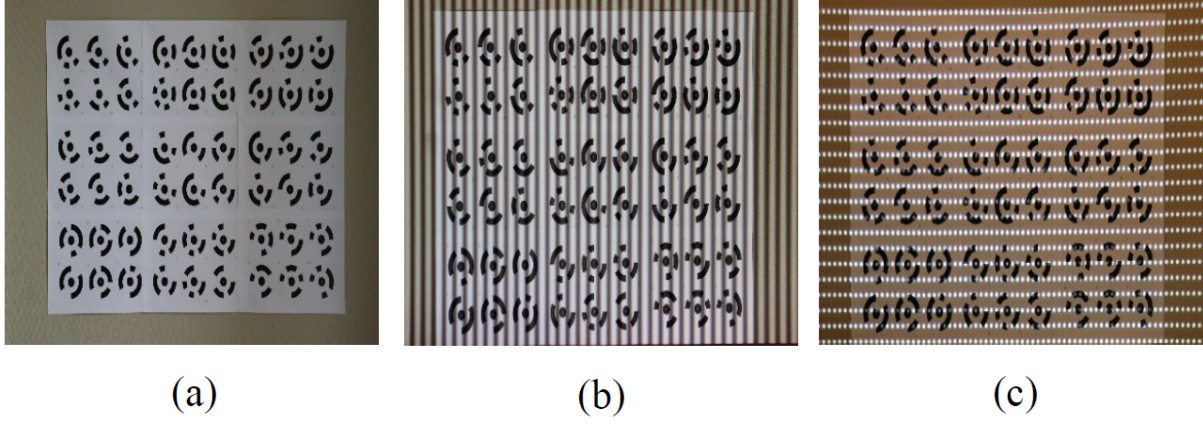

(a)  (b)  (c)

**Figure 4.** Illumination conditions. (**a**) Normal illumination; (**b**) striped structured light; (**c**) lattice-structured light.

*3.2. Evaluation Indicators*

We assume that the total number of CCTs in a photo is $N$ ($N$ in this algorithm is known; see Section 2 for suggestions on setting). The number of targets identified by the identification method is $n$. After verification, the number of CCTs is indeed $I$. The false positive is $n - I$. Among the correctly identified $I$ target, the number of correct decodes is $d$.

Three indicators are defined to evaluate the performance of identification methods and to test the ability of this algorithm to improve the performance of identification methods.

$$Correct\ Rate\ C = \frac{I}{n} \tag{11}$$

$$Recall\ Rate\ R = \frac{I}{N} \tag{12}$$

$$Decoding\ Rate\ D = \frac{d}{N} \tag{13}$$

The correct rate $C$ reflects the ability of the identification method to identify correctly. The recall rate $R$ demonstrates the power of the identification method to detect CCTs, while a low recall rate $R$ indicates that some targets have not been identified. The decoding rate $D$ reflects the ability to decode effectively.

*3.3. Experimental Results*

3.3.1. AM and AHF-AM Identification Results

Figure 5a–c show the identification results of AM. Under normal illumination conditions, AM can perfectly identify all CCTs. However, under complex illumination conditions, the identification results of AM are not ideal, especially under striped structured light, and CCTs cannot be detected. Surprisingly, under lattice-structured light, AM can identify 18 targets. After verification, these 18 targets are CCTs without false positives.

Figure 5d,e shows the identification results of AHF-AM. After processing with the AHF algorithm, the image quality was significantly improved. Under striped structured light, AHF-AM can perfectly identify all CCTs. Under lattice-structured light, 42 CCTs were identified.

Table 3 summarizes the identification results of AM and AHF-AM; "-" in the table indicates that it has not been processed. Because under normal illumination conditions, the targets identified by AM are equal to the total number of targets, the AHF algorithm was not executed. We can see that AHF-AM has an absolute advantage in identifying CCTs under striped structured light. Compared with AM's identification results, the AHF-AM recall rate increased by 45% under lattice-structured light. Surprisingly, the AM identification method has no false positives and can accurately decode if a CCT is detected.

3.3.2. TI and AHF-TI Identification Results

Figure 6a–c show the identification results of TI. It is worth noting that the decoding display of TI and AM is different. AM is numbered through a lookup table after decoding, and TI is the displayed decoded value. The identification performance of TI could be better than AM's, and even under normal illumination conditions, it cannot fully identify CCTs, resulting in false positives. Under striped structured light, TI also cannot detect CCTs. Under lattice-structured light, up to 342 false positives are not correctly identified as CCTs.

Figure 6d–f show the identification results of AHF-TI. Table 4 summarizes the identification results of TI and AHF-TI. After processing with the AHF algorithm, the image quality was significantly improved. Under normal illumination conditions, although not all CCTs are identified, false positives are reduced. Under striped structured light, AHF also performs well, and AHF-TI has an absolute advantage over TI in identification results. Under lattice-structured light, AHF-TI significantly reduces false positives. Although the

accuracy and recall rates are low, the identification results compared to TI are also entirely satisfactory. Surprisingly, although TI's identification performance is poor, it can accurately decode as long as a CCT is detected.

**Table 3.** Identification result statistics of AM and AHF-AM.

| Type | Methods | $\gamma H$ | $\gamma L$ | $N$ | $n$ | $I$ | $d$ | $C$ | $R$ | $D$ |
|---|---|---|---|---|---|---|---|---|---|---|
| Normal | AM | - | - | 54 | 54 | 54 | 54 | 100% | 100% | 100% |
| | AHF-AM | - | - | - | - | - | - | - | - | - |
| Stripe | AM | - | - | 54 | 0 | 0 | 0 | 0 | 0 | 0 |
| | AHF-AM | 2.8 | 0.82 | 54 | 54 | 54 | 54 | 100% | 100% | 100% |
| Lattice | AM | - | - | 54 | 18 | 18 | 18 | 100% | 33% | 33% |
| | AHF-AM | 2.6 | 0.44 | 54 | 42 | 42 | 42 | 100% | 78% | 78% |

**Table 4.** Identification result statistics of TI and AHF-TI.

| Types | Methods | $\gamma H$ | $\gamma L$ | $N$ | $n$ | $I$ | $d$ | $C$ | $R$ | $D$ |
|---|---|---|---|---|---|---|---|---|---|---|
| Normal | TI | - | - | 54 | 65 | 35 | 35 | 54% | 65% | 65% |
| | AHF-TI | 3.9 | 0.93 | 54 | 58 | 36 | 36 | 62% | 67% | 67% |
| Stripe | TI | - | - | 54 | 0 | 0 | 0 | 0 | 0 | 0 |
| | AHF-TI | 3.8 | 0.70 | 54 | 58 | 45 | 45 | 78% | 83% | 83% |
| Lattice | TI | - | - | 54 | 342 | 0 | 0 | 0 | 0 | 0 |
| | AHF-TI | 3.7 | 0.66 | 54 | 47 | 15 | 15 | 32% | 28% | 28% |

Table 5 shows the comprehensive evaluation results of 50 AM, AHF-AM, TI, and AHF-TI images under normal illumination, striped, and lattice-structured light. Although the identification performance of AM and TI is different, as long as a CCT is detected, it can be accurately decoded, so the recall rate and decoding rate are the same. Under normal illumination conditions, AM performs excellently and can perfectly identify all CCTs. Under striped structured light, neither AM nor TI can detect CCTs. However, AHF performs exceptionally well in processing striped structured light. AHF-AM can correctly identify all CCTs without false positives. AHF-TI's identification under striped structured light is even better than that under normal illumination conditions. Under lattice-structured light, both AHF-AM and AHF-TI significantly improve identification results. In the identification statistics of TI and AHF-TI, the recall rate is higher than the correct rate, indicating the presence of false positives. The statistical results in Table 5 are similar to those in Tables 3 and 4 and reflect the combined identification of AM, TI, and AHF with high stability.

**Table 5.** Comprehensive evaluation of AHF algorithm.

| Methods | Normal | | | Stripe | | | Lattice | | |
|---|---|---|---|---|---|---|---|---|---|
| | $C$ | $R$ | $D$ | $C$ | $R$ | $D$ | $C$ | $R$ | $D$ |
| AM | 100% | 100% | 100% | 0 | 0 | 0 | 100% | 22% | 22% |
| AHF-AM | - | - | - | 100% | 100% | 100% | 100% | 78% | 78% |
| TI | 53% | 62% | 62% | 0 | 0 | 0 | 0 | 0 | 0 |
| AHF-TI | 63% | 68% | 68% | 80% | 83% | 83% | 35% | 26% | 26% |

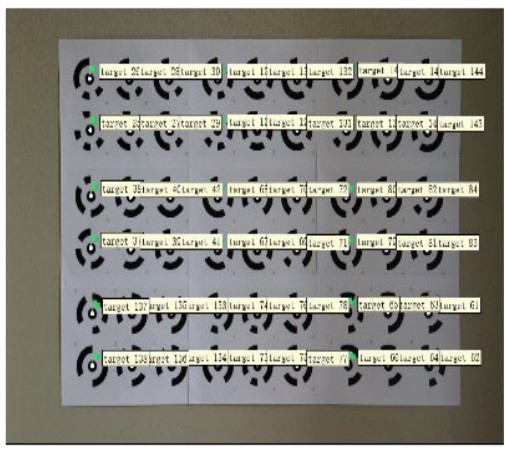

(a)

(b)

(d)

(c)

(e)

**Figure 5.** AM and AHF-AM identification results. (**a**) Identification of AM under normal lighting conditions; (**b**) identification of AM under striped structured light; (**c**) identification of AM under lattice-structured light; (**d**) identification of AHF-AM under striped lighting conditions; (**e**) identification of AHF-AM under lattice lighting conditions.

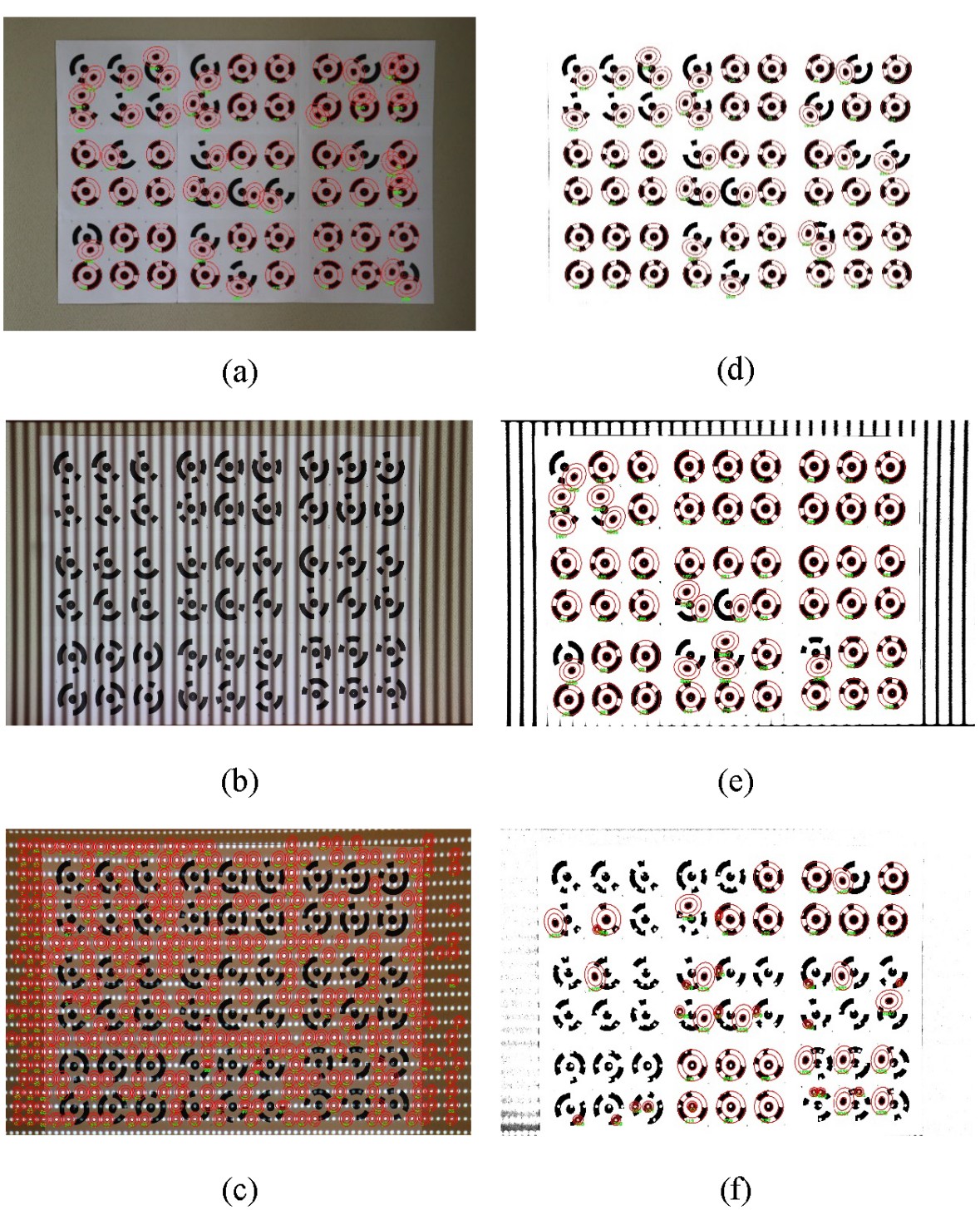

**Figure 6.** TI and AHF-TI identification results. (**a**) Identification of TI under normal lighting conditions; (**b**) identification of TI under striped structured light; (**c**) identification of TI under lattice-structured light; (**d**) identification of AHF-TI under normal lighting conditions; (**e**) identification of AHF-TI under striped structured lighting conditions; (**f**) identification of AHF-TI under lattice-structured lighting conditions.

### 3.3.3. The Precision of Center Positioning of CCTs

The precise identification of CCTs includes correct decoding and obtaining accurate center positioning coordinates. The center positioning accuracy of CCTs is of great significance for the accuracy, stability, and reliability of close-range photogrammetry. It affects point cloud reconstruction, camera positioning, feature matching, and measurement ac-

curacy evaluation. Therefore, it is necessary to evaluate the center positioning accuracy of CCT.

Through experiments, we found that AM and TI could not identify CCT at all under stripe-structured light, so it was impossible to evaluate the center positioning accuracy of CCTs. Under the lattice-structured light, there are a lot of false positives in TI identification; after verification, CCT is not correctly identified, and the AM identification effect is not ideal. AM is a very mature and robust commercial photogrammetric software. Through experiments, AM can accurately identify CCTs under normal lighting conditions. Therefore, we consider the CCT center positioning coordinates identified by AM under normal lighting conditions (AM-Normal represents the identification results of AM under normal lighting conditions) as ground truth to evaluate the impact of the AHF algorithm on the center positioning accuracy of CCT identification methods under complex lighting conditions. Evaluating the center positioning accuracy of CCTs that have yet to be correctly identified is meaningless. Therefore, on the identified CCTs benchmark, we calculate the offset RMSE of each method relative to the image coordinates of AM-Normal in the X direction, Y direction, and plane direction (XY direction). As shown in Table 6, "-" represents CCTs that were not correctly identified.

**Table 6.** Comparison of center positioning accuracy of various methods.

| Types | Methods | RMSE X (px) | RMSE Y (px) | RMSE XY (px) |
|---|---|---|---|---|
| Normal | TI | 0.312 | 0.214 | 0.378 |
| | AHF-TI | 0.201 | 0.170 | 0.263 |
| | AM | - | - | - |
| Stripe | AHF-AM | 0.070 | 0.093 | 0.116 |
| | TI | - | - | - |
| | AHF-TI | 0.189 | 0.159 | 0.247 |
| | AM | 0.125 | 0.203 | 0.238 |
| Lattice | AHF-AM | 0.091 | 0.103 | 0.137 |
| | TI | - | - | - |
| | AHF-TI | 0.256 | 0.219 | 0.337 |

AHF serves as an image preprocessing algorithm to optimize the image of CCTs under complex illumination conditions. From Table 5, we can see that the center positioning accuracy of AM is very robust. Whether under stripe- or lattice-structured light, the center positioning accuracy of AHF-AM is 0.07–0.1 px and is slightly worse under lattice-structured light. The performance of TI is poor, but according to Table 5, we can see that after AHF preprocessing, the center positioning accuracy of AHF-TI is between 0.2–0.4 px. Although it is a significant error for high-precision photogrammetry [28], it also significantly improves the center positioning accuracy of TI. Through experiments, it has been verified that our AHF algorithm significantly improves the center positioning accuracy of CCTs under complex lighting conditions.

## 4. Discussion

Through identification experiments on CCTs, we verified that HF can be applied to photogrammetry and can effectively cope with the impact of complex illumination. The proposed AHF overcomes the tedious process of determining HF parameters through repeated experiments. Regardless of which identification method is combined and which type of illumination conditions are applied, it can achieve universal and automatic characteristics.

### 4.1. Why Can Our AHF Be Combined with Any Identification Method?

Figure 2 shows the overall process of our algorithm. The core idea of the entire algorithm is that as an independent module, the identification method of CCTs is used to identify the number of CCTs $n$ in the image after HF and is used as the objective function of GA optimization for global optimization. The proposed AHF does not participate in

the work of the CCT identification method. Still, it only serves as the result of image preprocessing and calling the identification method in the optimization process to realize the automatic optimization of the image affected by complex illumination. Therefore, our AHF can be combined with any identification method, including various photogrammetric software or the identification method of CCTs proposed by scholars. In this article, we selected representative identification methods for experiments, namely, advanced commercial photogrammetric software Agisoft Metashape and the latest published traditional CCT identification method.

### 4.2. Analysis of Experimental Results

4.2.1. Experimental Results under Striped Structured Light

The original intention of our work was to overcome the influence of uneven illumination and shadow areas. To thoroughly verify the performance of the proposed algorithm, we set rigorous experimental conditions to cast stripe-structured light on the measured image. This case covers many densely shaded areas and the impact of uneven lighting.

Not surprisingly, such strict experimental conditions posed obstacles to the identification method of CCTs. AM and TI cannot work effectively under stripe-structured light. Observing the image of the CCT under the stripe-structured light, the dense shadow areas are distributed on the coding ring band and the center point of the CCT. The identification method of CCTs based on grayscale statistics is ineffective for images with frequent grayscale changes.

In our cognition, the image's stripes (structured light illumination part) have a spatial frequency similar to that of the CCT. Therefore, some of the CCT patterns will be filtered out by HF. However, what surprised us was that our AHF performed too flawlessly, as shown in Figures 5d and 6e and by the statistics in Tables 2–4. AM could identify correctly 100% of the time, while TI's identification results were equally excellent, even exceeding the identification results under normal lighting conditions. We did not apply a binarization method with appropriate thresholds to the image and only processed it through HF.

By observing the identification images in Figures 5d and 6e, we can observe that the same stripe patterns are filtered out in the white paper area (CCTs area) but remain outside that area. We analyze based on the illuminance–reflection model of the image, and image $f(x, y)$ can be represented as the product of the illuminance field (illumination function) $i(x, y)$ of the light source and the reflection field (reflection function) $r(x, y)$ of the reflected light of the object in the scene, i.e.,

$$f(x, y) = i(x, y) \cdot r(x, y) \tag{14}$$

Equation (14) is generally called the illuminance–reflection model of an image. The illumination function describes the illumination of a scene, and its properties depend on the light source and are independent of the scene. The reflection function describes the content of a scene, and its properties depend on the characteristics of the imaging object, independent of lighting. Due to the slow variation of lighting brightness, the spectrum of the lighting function is concentrated in the low-frequency range. Due to the rapid spatial variation of the reflection function with different image details (such as edge parts of objects), the spectrum of the reflection function is concentrated in the high-frequency range. In this way, the image can be understood as the result of the product of the high-frequency and low-frequency components according to Equation (14). After Fourier transform, the middle part of the Fourier spectrum is the low-frequency part, and the higher the outer edge frequency, the higher the frequency. The edges in the image correspond to high-frequency components, so image sharpening can be achieved using a high-pass filter.

As shown in Figure 7, the CCT image (stripe illumination part) is processed by HF with different parameter values. Figure 7a–c show the HF effect with a fixed $\gamma H$ value and an increase in $\gamma L$ value. We found that as the $\gamma L$-value increases, the overall brightness and sharpness of the image are improved, which helps to identify CCTs. This seems to contradict the original intention of HF design (suppressing illumination components). However, it is

precise that the illumination component is increased because of the increase of the $\gamma L$ value, which leads to the narrowing of stripes and reduces the influence of stripe-structured light. Figure 7d,e shows the HF effect with fixed $\gamma L$-value and increased $\gamma H$-value. The $\gamma H$-value is used to adjust the high-frequency reflection component. The improvement of the $\gamma H$-value highlights the contour details of the CCT. Therefore, it can be seen that with the increase of the $\gamma H$-value, the overall brightness and sharpness of the image are significantly improved, and the stripe width also changes with the change of the $\gamma H$-value. Both $\gamma H$ and $\gamma L$ values affect the overall brightness and sharpness of the image, and both $\gamma H$ and $\gamma L$ values jointly affect the filtering effect of the circular encoded target image. Therefore, adjusting only one value of $\gamma H$ and $\gamma L$ is not feasible.

Figure 8 shows the impact of HF with different parameter values on CCT identification. Indeed, the identification effect also varies with the combination of $\gamma H$ and $\gamma L$ values. When $\gamma H$ = 2.0 and $\gamma L$ = 0.05 are combined, the image is too dark, which affects identification. Combining $\gamma H$ and $\gamma L$ values in Figure 8d–f improves the image's overall brightness. However, the stripes are still distributed on the CCT, resulting in poor identification performance. Figure 8c,f,i show the effect when both $\gamma H$ and $\gamma L$ values are high. The image brightness is high, and the sharpness is also high. However, excessive enhancement of image edge details results in incomplete CCT bands, which affects the identification of CCTs. This also explains the necessity of designing automatic optimization HF algorithms.

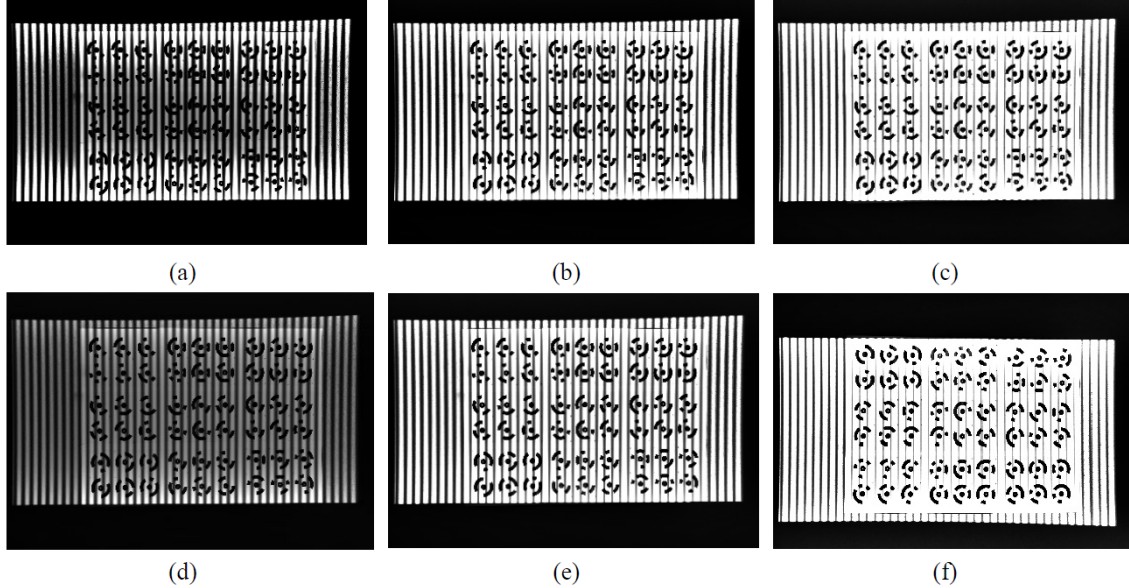

**Figure 7.** Processing effect of HF with different parameter values on CCT illuminated by stripe-structured light. (**a**) $\gamma H$ = 2.2, $\gamma L$ = 0.25; (**b**) $\gamma H$ = 2.2, $\gamma L$ = 0.55; (**c**) $\gamma H$ = 2.2, $\gamma L$ = 0.85; (**d**) $\gamma H$ = 1.8, $\gamma L$ = 0.85; (**e**) $\gamma H$ = 2.0, $\gamma L$ = 0.85; (**f**) $\gamma H$ = 2.3, $\gamma L$ = 0.85.

HF with appropriate parameter values can filter out all interference in shaded areas, stretch image contrast, smooth the image, and highlight the details of the image contour. CCTs on white paper enable HF to compress the image's dynamic range under high contrast in densely shaded areas, resulting in sharper CCT images. This also explains why AHF-TI's identification results under stripe results exceed those under normal lighting and why stripe areas are still outside the CCT area.

Our AHF can help us find the most suitable combination of HF parameter values for this identification method. Therefore, AHF-AM can identify perfectly under the stripe-structured light. Its optimization stop standard is that the number of identified targets $n$ equals the total number of targets $N$. The optimization stop criterion for AHF-TI is to reach the maximum optimization generation.

### 4.2.2. Experimental Results under Lattice-Structured Light

Identifying CCTs under the projected lattice-structured light can simulate the illumination of points, multiple light sources, and local highlight, and the experimental conditions are still strict enough such that such a dense lattice-structured light shines on the measured image to form a dense oval spot, which poses a considerable obstacle to the identification of CCTs.

The dense spots formed under the lattice-structured light significantly affect the identification of TI. TI mistakenly identifies the light spot as a CCT, and there are many false positives. Although the identification results of AM were poor, there were no false positives. By observing the CCTs identified by AM, we found a standard feature: the central positioning point of the CCT and the absence of light spot distribution on the central ring of the coding band. Even if many light spots are distributed in other positions of the CCT, it does not affect identification. Therefore, we can infer the identification strategy of AM: detecting the central positioning point of the CCT, locating it to the center of the coding band based on its geometric structure, and then decoding it. This strategy leads to its identification failure under stripe-structured light. Obviously, under the lattice-structured light, the identification of AM is random.

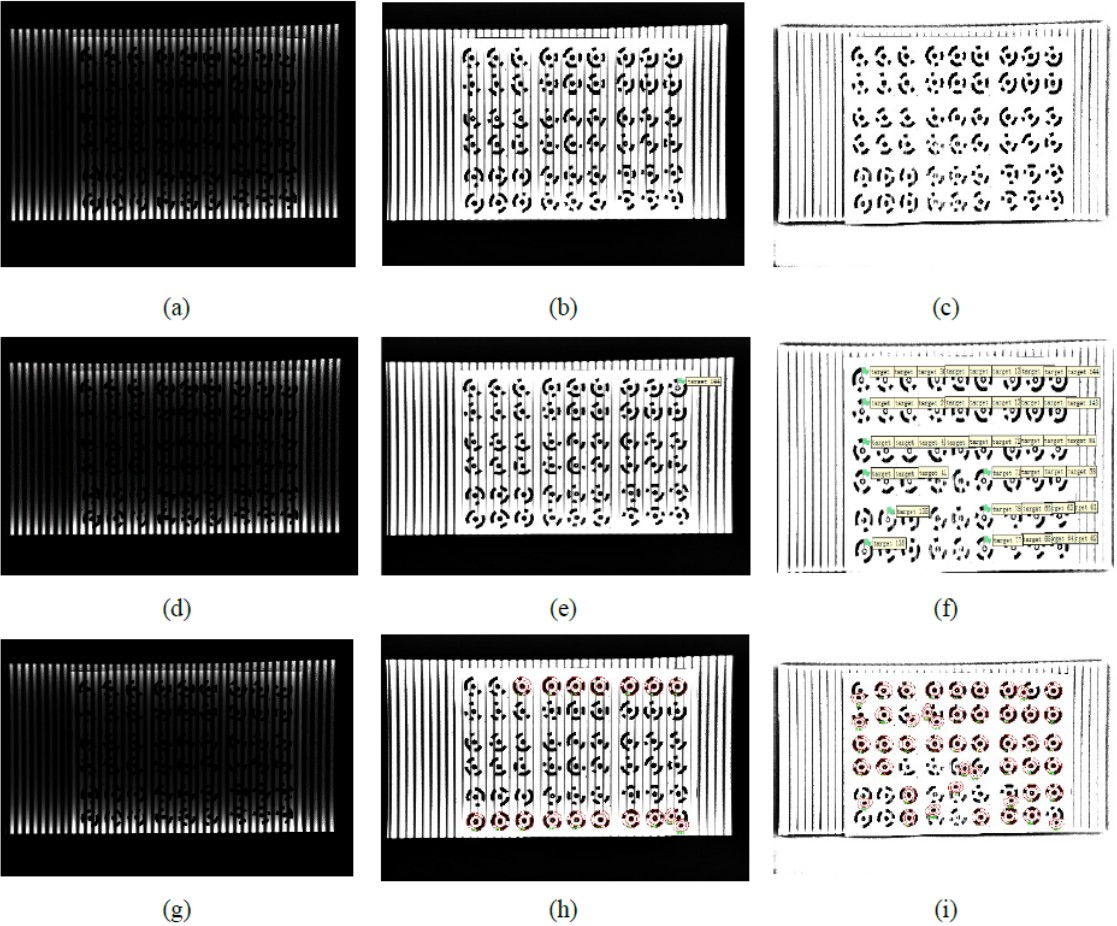

**Figure 8.** Effect of HF with different parameter values on CCT identification under stripe-structured light illumination. (**a**) Original image $\gamma H = 2.0$, $\gamma L = 0.05$; (**b**) original image $\gamma H = 2.5$, $\gamma L = 0.60$; (**c**) original image $\gamma H = 3.9$, $\gamma L = 0.85$; (**d**) AM's identification of (**a**); (**e**) AM's identification of (**b**); (**f**) AM's identification of (**c**); (**g**) TI's identification of (**a**) ; (**h**) TI's identification of (**b**); (**i**) TI's identification of (**c**).

Under the lattice-structured light, HF is still good for optimizing the image affected by illumination and significantly improves the identification results of the identification

method. By observing Figures 5e and 6f, HF can effectively filter out light spots. Through the observation in Figure 9, we find that the combined effect of HF parameters is similar to that under the illumination of stripe-structured light. Under the combination of low $\gamma H$ and $\gamma L$ values, the image is generally darker; however, the impact is more severe under lattice-structured light. With the adjustment of the $\gamma H$ and $\gamma L$ values, the image brightness and sharpness are improved, but the light spot on the CCT causes the coding ring band to be incomplete. Therefore, filtering the light spots distributed on the coding ring involves a trade-off between filtering performance and identification results.

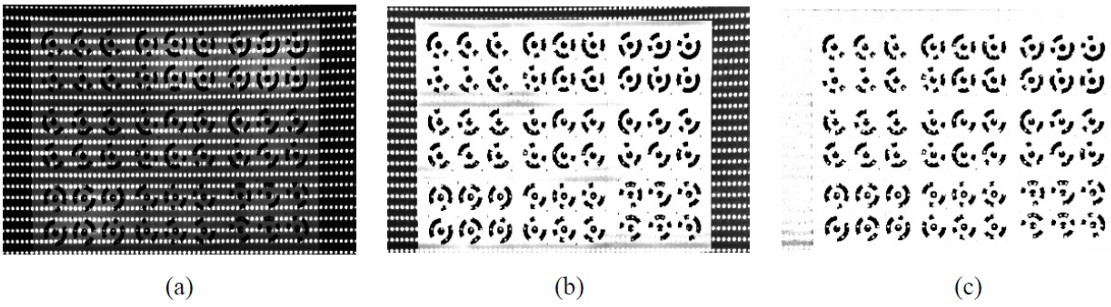

(a)                                            (b)                                            (c)

**Figure 9.** The processing effect of HF with different parameter values on CCTs illuminated by lattice-structured light. (**a**) $\gamma H$ = 2.4, $\gamma L$ = 0.32; (**b**) $\gamma H$ = 2.6, $\gamma L$ = 0.50; (**c**) $\gamma H$ = 2.8, $\gamma L$ = 0.80.

The AHF algorithm automatically finds this identification method's most suitable combination of HF parameters. The statistical results in Figure 6c,f and Table 3 show that the AHF algorithm reduces many false positives in TI identification. Compared with the identification of AM and TI, the identification results of AHF-AM and AHF-TI under lattice-structured light have comparative advantages.

### 4.3. Efficiency Issues

The complexity analysis of the GA depends on various factors, such as population size $n$, individual size $m$, generation $g$, and the fitness function used. In the worst case, it has O ($gnm$) time complexity [61]. Due to the frequent use of CCT identification methods in our algorithm, the efficiency of this algorithm mainly depends on the efficiency of the identification method.

The processing efficiency of TI is relatively low, and the processing time varies depending on different image types. For example, TI mistakenly identifies many spots of lattice-structured light as targets, and it takes an average of 3 s per image to process such images. However, the processing efficiency of AM is exceptionally high, with a total consumption of approximately 1 s for processing nearly 1000 different types of images. Therefore, combining our proposed AHF algorithm with photogrammetric software will have higher processing efficiency and be used for practical industrial applications.

We sacrifice time costs to identify CCTs under complex illumination conditions accurately. For close-range photogrammetry tasks, the primary working mode is offline [62], so the algorithm proposed in this article can meet the requirements.

### 4.4. Application Prospects

4.4.1. The Application of AHF in Photogrammetry in the Field of Remote Sensing

CCTs are essential in photogrammetry, providing accurate position, pose, and scale information for measurement tasks through feature extraction, positioning and navigation, calibration and correction, 3D reconstruction, and data registration [63]. It is worth noting that the SFM technology with no targets still dominates in unmanned aerial vehicle (UAV) photogrammetry, but CCTs are widely used as ground control points [64]. The contributions and potential applications of the proposed algorithm to photogrammetry in the field of remote sensing are as follows:

1. Robustness to Constantly Changing Lighting Conditions:
   Photogrammetry typically involves capturing images in outdoor environments with significant changes in lighting conditions [65]. The ability to identify CCTs under complex lighting conditions, including shadows, highlights, and uneven lighting, ensures the consistency and robustness of the photogrammetric process. It reduces the probability of CCT identification failure or errors due to challenging lighting conditions.

2. Improving Reconstruction Accuracy:
   Accurately reconstructing 3D models and maps is the primary goal of UAV photogrammetry. Accurate identification of CCTs under complex lighting conditions improves the accuracy of the reconstruction model. It ensures reliable correspondence between images and facilitates more accurate triangulation and point cloud generation, thereby achieving higher-quality 3D reconstruction.

3. Multifunctional Application:
   It successfully identifies CCTs under complex lighting conditions, expanding the applicability of UAV photogrammetry in various scenarios. It can capture high-quality data in challenging lighting environments, enhancing the effectiveness of data collection and analysis based on UAV photogrammetry.

### 4.4.2. AHF for Close-Range Photogrammetry and Structured Light Fusion

Our team's previous work [5] research found that by combining close-range photogrammetry and structured light technology, combining their respective advantages, there is excellent potential for high-precision measurement of the three-dimensional topography of large objects [1,66,67]. Close-range photogrammetry technology captures high-resolution images from multiple angles, integrates global control points, and can accurately reconstruct the geometric structure of objects. Structured light technology provides accurate depth measurement to obtain local high-density point clouds [68]. This combination can significantly improve the accuracy of depth maps and surface reconstruction. Through experiments, it is verified that the algorithm can perfectly identify the CCT under the stripe-structured light, which provides a new idea and potential application for the technical fusion of the two.

Complex lighting conditions may pose challenges in capturing accurate surface information. However, if the CCT can be ideally identified under these conditions, close-range photogrammetry and structured light technology integration will become more robust. The CCT can be excellently identified in the stripe-structured light image, which means that the CCT can still be used as the tie point between the images to achieve accurate and reliable point correspondence, improving the accuracy of the matching process and better integrating the two technologies' data. The proposed algorithm can ensure accurate alignment, matching, and calibration in challenging lighting scenes.

### 5. Conclusions

This article proposes an AHF algorithm as a preprocessing technique combined with a CCT identification method to address the impact of complex illumination. We set rigorous experimental conditions, casting stripes and lattice-structured light on the measured image, covering the complex lighting conditions of the point light source, multiple light sources, uneven lighting, and shadow areas. Experiments were conducted using commercial photogrammetric software AM and the state-of-the-art traditional CCT identification method, TI.

The experimental results show that although the current identification methods for CCTs are pretty mature, they still fail under complex illumination conditions. We find that HF with appropriate parameter values can effectively filter out the influence of stripe-structured light illumination and has satisfactory results for lattice-structured light illumination. Therefore, HF has excellent potential in dealing with image problems affected by complex illumination conditions. The AHF proposed in this article can achieve automatic

parameter adjustment and can be combined with any CCT identification method with automatic and universal features, that is, users only need to input photos without any other operations.

The future application prospects of this algorithm: As a preprocessing technology, it can be integrated into photogrammetric software. Alternatively, scholars can use it to develop more robust CCT identification methods without considering the impact of complex illumination. This algorithm also provides a new idea and potential solution for the fusion of close-range photogrammetry and stripe-structured light technology. Our team's next work plan is to apply this algorithm to the fusion application of close-range photogrammetry and structured light technology.

**Author Contributions:** Conceptualization, H.S. and C.L.; methodology, H.S. and C.L.; formal analysis, H.S. and C.L.; investigation, H.S. and C.L.; data curation, H.S. and C.L.; writing—original draft preparation, H.S. and C.L.; writing—review and editing, H.S. and C.L. All authors have read and agreed to the published version of the manuscript.

**Funding:** This research received no external funding.

**Data Availability Statement:** Data supporting the findings of this study are available from the authors upon request.

**Conflicts of Interest:** The authors declare no conflict of interest.

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
