# Peer review of "AHF: An Automatic and Universal Image Preprocessing Algorithm for Circular-Coded Targets Identification in Close-Range Photogrammetry under Complex Illumination Conditions"

_remotesensing, doi:10.3390/rs15123151_

Round 1

Reviewer 1 Report

The abstract consists of 263 words - more than recommended. The text seems too long, too detailed. Some sentences are written in a difficult way. However, this does not reduce the quality of the described studies.

It would be good to do one more practical test, e.g. in the shade of trees, etc., and thus demonstrate the practical effectiveness of the method.

Author Response

Thank you very much for the reviewer's suggestions and comments.

Point 1: The abstract consists of 263 words - more than recommended.

Response 1: We noticed that the recommended abstract is limited to 250 words, so we focused on this point when writing the manuscript. The abstract in the manuscript has a word count of 242 words. The 263 characters noticed by the reviewer include the numbering after each line for the convenience of reviewing the manuscript. If these numbers are included, it is indeed 263 characters. But these numbers will not appear after finalization, so our article abstract has a word count 242. Thank you for your careful review.

Point 2:It would be good to do one more practical test, e.g., in the shade of trees, etc., and thus demonstrate the practical effectiveness of the method.

Response 2: We are very fortunate to have shared ideas and perspectives with the reviewers. Before the text was finalized, experiments were added, including experiments in shaded areas under trees, and even included conditions such as abundant sunlight, insufficient sunlight, cloudy days, and camera exposure. However, we ultimately removed these experiments when the article was finalized. The reason is that the purpose of such measured conditions is to create shaded areas, uneven lighting, etc. However, it must highlight the complexity, making it challenging to call complex lighting conditions. It is better to cast stripes or lattice Structured light for experiments directly. Such experimental conditions are very complex, including the shadow areas we need (more complex than the shadow areas under the tree shade, and the shadow areas under the tree shade are relatively single), uneven lighting, multi-point light sources, etc. Therefore, we only retain the experimental conditions under the fringe and lattice Structured light in the manuscript, so these sufficiently cover the recognition requirements for CCT under the complex lighting conditions we need.

Thank you again for the valuable suggestions and comments from the reviewers. It is an honor to meet reviewers with ideas that align with us.

Reviewer 2 Report

Dear Authors,

The article is a study to examine the effect of uneven lighting on image quality. The AHF proposed in the article to study the effect of uneven illumination can perform automatic parameter tuning and be combined with any CCT identification method with automatic and universal characteristics.   The method using a GA is proposed to optimize the HF parameters using the identification results of the CCT as the objective function. It is stated that when HF is applied with appropriate parameter values, image details can be emphasized better, contributing to the improvement of photo quality and accuracy. In the article, Agisoft Metashape and the most recently published traditional CCT identification method were selected for experiments.

In the Introduction section, the purpose of the publication is given clearly. The relevant section does not contain unnecessary details that do not directly contribute to the research problem or objectives. Required literature summary about the publication is given. In the second part, the proposed algorithm is given clearly. The application part is also given in detail. However, the "Results" heading in row 271 can be changed to "Experiments and analysis". The algorithm given in the second section is tested in detail in this section. The analyzes and their results are given in detail. In the fourth chapter, the originality and effective aspects of the study are emphasized by making references to other studies on the subject. The positive aspects of the methodology are emphasized.

In summary, why the study was carried out, what it aimed for, and its contribution to originality with different studies were clearly explained. I think that the work is suitable for publication as it is.

Kind regards

Author Response

Thank you very much for the reviewer's comments and suggestions. Thank you again for the reviewer's affirmation of our article.

For line 271, pointed out by the reviewer, which is the title of section 3 in the article, we believe that the reviewer's suggestion is sufficient, and we have changed it to "Experiments and Analysis" according to the reviewer's opinion.

Thank you again for taking the time to review our article.

Reviewer 3 Report

In this article, authors have proposed an automatic and universal image preprocessing algorithm for circular-coded targets identification in close-range photogrammetry.

Comments/Suggestions

·         Figure 2 needs to be improved; the text in the boxes is overflowing

·         It is suggested to add a table for the specifications of the camera instead of text.

·         Many relevant papers have not been added to the literature. Some of them are the following:

1.       Forero, M.G.; Mambuscay, C.L.; Monroy, M.F.; Miranda, S.L.; Méndez, D.; Valencia, M.O.; Gomez Selvaraj, M. Comparative Analysis of Detectors and Feature Descriptors for Multispectral Image Matching in Rice Crops. Plants 202110, 1791. https://doi.org/10.3390/plants10091791

2.       Sharma, S.K.; Jain, K.; Shukla, A.K. A Comparative Analysis of Feature Detectors and Descriptors for Image Stitching. Appl. Sci. 202313, 6015. https://doi.org/10.3390/app13106015

3.       Lin Chen, Franz Rottensteiner & Christian Heipke (2021) Feature detection and description for image matching: from hand-crafted design to deep learning, Geo-spatial Information Science, 24:1, 58-74, DOI: 10.1080/10095020.2020.1843376

4.       https://www.isprs.org/proceedings/XXXVI/part3/singlepapers/P_05.pdf

Author Response

Thank you very much for the reviewer's comments and suggestions.

Point 1: Figure 2 needs to be improved; the text in the boxes is overflowing.

Response 1: We have rechecked the manuscript and found that Figure 2 needs to be changed. Thank you for the reviewer's careful pointing; we would greatly appreciate it.

Point 2: Adding a table for the camera specifications is suggested instead of text.

Response 2: Indeed, the opinions provided by the reviewer can provide readers with a more intuitive understanding of our experimental setup and are also more in line with the standards of paper writing. Based on the reviewer's opinion, we have set the camera specifications as a table.

Point3:Many relevant papers have not been added to the literature.

Response 3: Thank you for pointing out that we have incorporated the literature listed by the reviewer into the article.

Thank you again for the reviewer's comments and opinions.

Reviewer 4 Report

Fundamentally, the paper is OK, however the potential practically of the new approach to accurately measuring circular coded targets in complex lighting conditions in close-range photogrammetry is somewhat overstated. This is simply because a large part of project design in industrial & engineering photogrammetry centers upon AVOIDING complex lighting conditions, so the problem does not frequently arise. Hence the use of strobes, retroreflective targets, special lens filtering etc. It's conceivable that the approach might be useful in scanning situations of striped- & lattice-structure light, but then common practice in such circumstances is to measure the framework of coded targets in advance of the scanning, which partially alleviates the problem, and certainly the image-to-image matching problem.

Some specific shortcomings of the approach and paper are:

1) Much seems to depend upon all coded targets being present in an image. However, this is very rarely the case in standard, automated off-line & on-line photogrammetry. Coded targets are used to solve the image point correspondence problem, and it is not necessary to see a large number - or all - in a single image from a multi-station network. A complex 3D survey might involve 100+ targets & tens of images, but the number of codes per image may be a few as, say, 6 - 8. Any method which is contingent upon each image seeing most of the coded targets has virtually no chance of practical adoption.

2) There is no explicit analysis of the added computation time, though there  is vague reference to the algorithm requiring a few seconds per image. If this is the case, the process is unacceptably slow! More analysis is warranted.

3) Also the accuracy seems poorer than acceptable, with image centroids having a variability of a few 10ths of a pixel, rather than sub 0.1 pixel. Further analysis here would be beneficial.

4) There's a late focus upon UAVs, but it is quite uncommon for UAV networks to utilise coded targets (a few fledgling indoor experiments being exceptions); instead targetless SfM techniques dominate. This fact should be mentioned & the UAV-focussed section shortened.

Bottom Line: A reasonable, though somewhat esoteric contribution, with limited practical utility.

OK

Author Response

Thank you very much for the reviewer's suggestions and comments.

The reviewer's comments are very objective, and we strongly agree with the reviewer’s viewpoint. A large part of the projects in industrial photogrammetry are indeed processed by avoiding complex lighting conditions. Alternatively, processing methods such as strobes, retroreflective targets, and special lens filtering were used. However, our method also provides another solution for complex lighting conditions. In unavoidable industrial environments, our method can be helpful. Our method uses printed paper CCTs, a low-cost measurement solution, instead of retroreflective targets.

Point 1: Much seems to depend upon all coded targets being present in an image. However, this is very rarely the case in standard, automated off-line & on-line photogrammetry. Coded targets are used to solve the image point correspondence problem, and it is not necessary to see a large number - or all - in a single image from a multi-station network. A complex 3D survey might involve 100+ targets & tens of images, but the number of codes per image may be a few as, say, 6 - 8. Any method which is contingent upon each image seeing most of the coded targets has virtually no chance of practical adoption.

Response 1: Obviously, the reviewer is an expert in close-range photogrammetry and is very familiar with the process of close-range photogrammetry. Our method does not depend on all coded targets on the image. Densely arranging numerous coded targets on the image is impractical in practical applications. This article does this to verify our experiment's ability to identify coded targets under such complex lighting conditions. The primary purpose is to conduct comparative experiments to highlight the advantages of our algorithm, such as other methods that cannot identify or incorrectly identify. The coded target is printed out, a single coded target printed on white paper. Scattered individual coded targets are also used for the result analysis discussed in Section 4.

Point 2: There is no explicit analysis of the added computation time, though there is vague reference to the algorithm requiring a few seconds per image. If this is the case, the process is unacceptably slow! More analysis is warranted.

Response 2:

We propose a preprocessing algorithm for CCT recognition under complex lighting conditions rather than studying how to recognize CCTs. Because commercial close-range photogrammetry software nowadays can ideally recognize CCTs, such as the Agisoft Metapaper used in the article, which has superior performance and extremely fast processing speed. Processing 1000 images containing CCTs in the actual measurement data takes almost 1 second. However, the recognition effect of such mature Commercial software under complex lighting conditions is not good, so our algorithm was designed based on this, and the performance of the recognition method was optimized.

To highlight the general characteristics of our algorithm, a comparative experiment is added; that is, the latest published CCT recognition method is selected. Obviously, as the experiment in the manuscript, this recognition method has poor performance and low recognition efficiency. It takes a long time to process images under lattice Structured light, reaching 3 seconds per image, and there are many false positives. This is a drawback of this CCT recognition method, not a problem with our algorithm. Our algorithm cannot improve the processing speed of this CCT recognition method, but it does optimize its performance. Through such comparative experiments, the advantages of our proposed algorithm have been verified.

Our algorithm is combined with mature commercial photography software in practical applications.

Point 3: Also, the accuracy seems poorer than acceptable, with image centroids having a variability of a few 10ths of a pixel, rather than sub 0.1 pixel. The further analysis here would be beneficial.

Response 3: Thank you for the reviewer's correction. As in the second reply, our algorithm is based on image optimization, allowing the CCT recognition method to have the best recognition performance without being affected by harsh imaging conditions under complex lighting. We chose the commercial photogrammetric software Agisoft Metamask as the basic fact because we found that the difference of RMSE of center positioning accuracy after AM recognition is less than 0.1 pixel after image optimization under fringe Structured light. The latest published recognition methods have poor performance, and although the optimized positioning accuracy is still unacceptable, there has also been a significant improvement.

Point 4: There's a late focus on UAVs, but it is quite uncommon for UAV networks to utilise coded targets (a few fledgling indoor experiments being exceptions); instead targetless SfM techniques dominate. This fact should be mentioned & the UAV-focussed section shortened.

Response 4: Indeed, as commented by the reviewer, the SfM technology with no targets is mainstream in UAV networks. We have also noticed some scholars who hope to introduce coding targets in UAV networks. Therefore, we hope to provide them with new ideas and potential solutions under complex lighting conditions. We plan to quote the reviewer's viewpoint in the UAV discussion section.

Thank you again for the reviewer's suggestions and comments. The reviewer has also provided us with a new perspective. Thank you very much.